# Who Is Doing What in Home Care Services?

**DOI:** 10.3390/ijerph181910504

**Published:** 2021-10-07

**Authors:** Berit Irene Helgheim, Birgithe Sandbaek

**Affiliations:** Faculty of Logistics, Molde University College, 6410 Molde, Norway; birgithe.e.sandbak@himolde.no

**Keywords:** home care, staffing, health care organizations and systems

## Abstract

(1) Background: This paper investigates the distribution of work hours by activity, for the main staff categories in home care services in three rural Norwegian municipalities. In Norway these categories are registered nurses, assistant nurses and assistant health workers. (2) Methods: The three categories of home care staff recorded 20,964 eligible observations over 8 weeks. We identified 19 activities, which were recorded. The majority of staff used a smartphone application for the time measuring, while some staff used a manual form for reporting purposes. (3) Results: The registered nurses (RNs) spent 32% of their time on direct patient work, while driving accounted for 18%. Direct patient work and driving accounted for the majority of activities performed by assistant nurses (48% and 29%, respectively) and assistant health workers (70% and 17%, respectively). (4) Conclusions: The demand for home care services is increasing in terms of both size and complexity. Simultaneously, there is a growing shortage of skilled staff. RNs is the group with the least face-to-face time with patients. To meet the patients’ needs, it is necessary to discuss and modify existing home care service systems in order to use resources appropriately and effectively.

## 1. Introduction

Many countries in Europe have reported a shortage of staff associated with home care services [1,2]. Norway is confronted with the problem of recruiting sufficient numbers of educated staff. In 2015, the country reported a lack of 2350 registered nurses (RNs). In addition, during the same year, 12% of RNs in home care services took sick leave [3]. Statistics Norway has reported that in 2030, an under-coverage of 41,500 ANs and 13,000 RNs is expected [4]. 

Nursing staff in Norway are mostly women, so only a small percentage (about 10%) are men. The sector suffers from long-term sick leave and vacant positions. There is an approximately 18% shortage of RNs in home care and nursing homes, either because of vacant positions or long-term sick leave. At the municipality level other staff substituted for these missing RNs. Of these substitutes, about 30% had health-related education from homecare, 30% held a bachelor’s degree in a health-related subject, 15% were nursing students and 22% were employees without any formal education [5].

The payment of employees for health care workers in the municipalities in Norway is centrally determined. The level of payment is defined in the central regulation based on education level and seniority. Regardless of education level, the payment includes all types of tasks performed, including driving to and from homes. In addition, in the last 10 years, there has been a large and systematic discrepancy between planned and actual staffing in nursing homes and home nursing [5]. This is due to short-term sick leave, and those substituting for absences are most commonly ANs and AHWs [5]. 

This calls for a discussion on who is going to do what, meaning what is the best way of utilizing educated staff in home care and what is the present situation.

Home care services in Norway, which are part of the public health system, is free of charge. Each municipality must ensure that the inhabitants are given their legal right to have the care they need regardless of age, gender or socioeconomic status [6]. Patients must apply for home care services and a medical board decides, based on each patient’s medical condition, how many hours of care per day/week that patient will receive. Recently, the government has also become focused on having unpaid caregivers to provide social care. These are organized through humanitarian organizations, such as the Red Cross. 

The distribution of patients receiving home care services has changed in the last decade. In Norway, younger patient groups have seen a significant increase in numbers. The share of receivers under 67 years of age has increased by 10% since 2007 and now accounts now for 42% of all those receiving home care [7]. 

Simultaneously, patients are discharged from hospitals earlier, and further medical treatment and follow up is transferred to municipalities [8]. Registered nurses (RNs) working in home care report an increase in patients from hospitals, who are more complex and have complex needs, without staff being given any additional training or education. In the patient population, approximately 63 percent of patients have a moderate or extensive need for medical nursing at home [7], and those in the youngest and the oldest patient groups most commonly require moderate or extensive care.

Given the simultaneous increase in demand for home care services and the growing complexity of the conditions experienced by home care patients, this calls for novel and innovative approaches to organizing and delivering home care services.

Research into home care services has investigated how home teams work together across different aspects. Several researchers have focused on team efficiency and investigated how collaborative characteristics influence patient satisfaction or quality of care [9,10,11,12,13,14,15]. Other researchers have examined the impact of knowledge and the performance of medical procedures performed by professional caregivers [16,17,18].

As home care services vary by country, researchers conducted a cross-sectional investigation among six European countries, in which they investigated thirty-six home care organizations. This research identified and characterized six home-help models, some of which may describe best practices [15].

In operations management, home care service researchers have focused on optimizing resources in order to improve the systems. Investigations show how technology may have a positive impact on increasing efficiency [16], while other researchers have investigated staff scheduling models and route planning [17,18,19]. The assumption for these models is a certain distribution of the numbers of staff and patients. However, countries with a more comprehensive primary care may also have a wider variety of professions [20].

In Norway, formal caregivers are a part of the public system and are divided into three main groups: assistants, who have no formal education (AHWs); ANs, who have at least three years of high school education; and RNs, who have at least three years of education at the college or university level, meaning that they have achieved a bachelor’s degree. Therefore, the tasks and responsibility of each group differ.

The teams are set up primarily based on the geographical distribution of patients and the number of hours allocated by the medical board. Primarily, the teams consist of at least one RN, who is the medical professional responsible for the team, and the ANs and AHWs are distributed depending on the number of patients in a geographic district. When there is a lack of RNs, the ANs and AHWs will perform ordinary RN tasks to some extent. For example, AHWs can give medicine to a patient if they are authorized for this purpose. Some municipalities have training sessions after which they authorize AHWs to give medicine to patients. One issue in home care is that there are no clear regulations on who is allowed to do what kinds of medical procedures [21]. For example, a RN can delegate tasks to ANs or AWHs, but exactly what kinds of medical tasks one may delegate is a grey area.

Undergraduate training programs for RNs are equally balanced between theoretical subjects and practical training. The RN program includes subjects such as communication, daily life functions, pharmacology, anatomy and physiology, social science, nursing, pharmacology, pathology and microbiology, pedagogy, ethics and administration. The training focuses on integrating these theoretical subjects with practical experience.

The ANs, who have completed a high school education, are trained as general health care workers. Their program focuses on two main subjects, namely health promotion work and communication. Their education emphasizes basic nursing and activity therapist work. Their training teaches ANs how to handle and help patients with various illnesses and injuries in addition to providing instruction in communication, prevention, and ethics related to patient care. The program also combines practical training and theoretical subjects.

In contrast to RNs and ANs, AHWs undergo no specific health care education. This group provides practical help, such as shopping, house cleaning, meal preparation and personal hygiene. Some municipalities have designed internal training programs for these workers, but these curricula do not provide a formal qualification. In general, AHWs are trained through individual supervision by an educated worker.

Many researchers have investigated the role of RNs and have examined the type of work that they perform [22,23]. Often, these investigations have been based on interviews with or questionnaires conducted among patients, nurses and/or managers in home care services. Unlike previous research, the present paper investigates the actual time spent on various tasks. Although the majority of caregivers recorded the timestamps associated with their activities using an application installed on their smartphones, some used a manual form to report their duration of their activities.

The purpose of this paper is to investigate the distribution of work hours by activity for the three main staff categories in home care. The results could be used as input to develop novel, innovative approaches to organizing and delivering home care services, with a focus on the process level. 

## 2. Materials and Methods

The data is recorded in three rural Norwegian municipalities. Home care staff prospectively recorded the time they spent on 19 home care activities. For the purpose of analysis, some of the remaining 17 activities were clustered into activity groups. Two activities involved direct patient work (practical help and nursing), and six activities involved non-direct work. The total population for the three municipalities was 23,940. Table 1 explains the individual activities and their corresponding clustered activity groups.

The primary staff categories were RNs, ANs and AHWs. Table 2 provides information on participating staff categories and work hours.

This study included work performed during the day shift, which was defined as weekdays from 7 a.m. to 3 p.m., a total of 6702.2 work hours were recorded over eight weeks. Each shift is divided into work teams. 

The work teams include one RN, who is the team manager. The ANs and AHWs are distributed based on the number of patients in each geographic district and the number of hours allocated per patient according to the medical board. 

ANs typically have 8–12 patients on a day shift, while AHWs have 8–16 patients. The number of patients per shift for RNs may vary because they need to cover for nursing shortages, which happens every week. Normally RNs will have between 6 and 10 patients on their list.

In addition to individual activities, the data included information on staff category and timestamps (i.e., date and start and end times) for each activity. Prior to the analysis, 399 (1.9%) observations were deleted due to coding errors, which resulted in 20,964 observations that were eligible for analysis. Coding errors were caused by activities running for more than 24 h due to users not indicating the end of an activity.

To record their activities, the majority of the staff used a time tracker application installed on their smartphones. However, some of the staff recorded their activities manually because they did not have access to a smartphone or preferred to use a manual registration form for other reasons. Prior to the project, staff participated in selecting, defining, and categorizing the specific activities that would be recorded. Furthermore, all staff were taught how to install and use the application. There was no significant difference in the distribution of the times for data collected electronically and manually.

The data was analyzed using Stata, version 15.1, StataCorp, Texas, USA. The Mann–Whitney test was used to calculate the statistical significance of the observed differences in the duration of the different activities performed by staff category. For statistical analyses, *p* < 0.05 was considered statistically significant. Appendix A reports the distribution of time (median, interquartile) and *p*-values.

## 3. Results

Over the 8 weeks examined, the staff carried out a total of 6702 h of work (Table 2), which consisted of 20,964 observations. Of the total work hours, 22%, 42% and 37% were performed by AHWs, ANs and RNs, respectively. These proportions reflect the composition of the participating staff categories. Significant differences are reflected in the proportions of work hours spent on different activities across the three staff categories (Table 3).

Nursing constituted 31% of the total work performed by RNs (median = 13 min), who also provided a small amount (1%) of practical help. Nursing constituted 42% of the total work performed by ANs (median = 13 min), while practical help accounted for 6% (median = 13 min). The AHWs primarily provided practical help, accounting for 50% (median = 75 min) of their total work, while nursing accounted for 20% (median = 15 min) of their work. The test between ANs and AHWs on time spent on practical work and nursing was significantly different (*p* = 0.00), although it was not significant between RNs and ANs. Despite the differences in proportions of nursing and practical help for ANs versus RNs (*p* = 0.00), the median times for these activities were similar for these two staff categories.

All three staff categories spent a substantial amount of time driving. These proportions amounted to 18% for AHWs (median = 8 min), 29% for ANs (median = 6 min), and 17% for RNs (median = 5 min; *p* = 0.00).

Reporting constituted 10% of RNs’ work, 12% of ANs’ work, and 4% of AHWs’ work (median = 30 min for all three categories; *p* = 0.04 between ANs and RNs).

The remaining proportions of the total work hours for RNs consisted of 22% administration (14% organizational administration and 8% patient administration), 6% documentation, 10% drug management, and 2% teaching activities.

## 4. Discussion

This paper employed prospectively recorded data to investigate the distribution of work hours by activity for the three main staff categories in home care services. The results indicate the existence of major differences in the proportions of work hours spent on different activities across the three staff categories. 

The RNs devoted 68% of their time to non-direct activities and, consequently, 32% to direct patient care. These findings differ from the findings of other studies, which have found that RNs’ main tasks are primarily related to direct work with patients [24,25]. Most of these works, however, have been based on qualitative data, such as that obtained through interviews and surveys. In contrast, we have prospectively measured the actual time spent on different activities.

Other researchers have also suggested that more of RNs’ time is allocated to those patients with the highest care needs [24,25]. In light of the increasing demand and more complex patients that have emerged in recent years, one might have expected that RNs would spend a greater proportion of their time on nursing activities. 

The discussion in Norway on how to solve the challenges home care services are facing, is not primarily connected to payment, because wages are considered to be fairly decent for all three groups. For RNs, there are other dimensions that are more important than the wages, such as shortage of trained employees that overloads nurses and the increasing complexity of home care patient conditions.

Driving is one of the most time-consuming tasks (18%). Some patients are living in small centers for the elderly. Patients in the same geographic clusters are normally handled by one team, and sometimes these teams can walk from one patient to the next without having to drive. However, all three municipalities in our investigation have major challenges because of how spread out the population is. As research in home care services has suggested, one may reduce the time spent on driving by using a more efficient route-planning algorithm [19]. However, this requires information such as the time window for visits, GPS coordinates and particular task information. These data were not available in our investigation.

The time spent on each visit varies according to the nature of the task. Nursing visits very often require a RN because of medical procedures such as wound management, rehabilitation, post-surgery follow up, intravenous fluid therapy and pain management for terminally ill patients. Some of the medical procedures that may be done by ANs are rehab training, less advanced wound management and administration of oral medication. AHWs are normally involved in more practical help and personal hygiene. One of the main problems is that patients do not get enough social care through home care services. The workload makes it difficult for workers to spend more than the allocated time with patients. If they use more time than what is expected, they have to either skip lunch or work overtime at the end of the day. The system today might be too tight: more flexibility is needed to deal with individual needs.

Administrative work accomplished by RNs typically includes updating care plans, contacting GPs, hospitals and relatives and documenting medication updates. They are also involved in ordering supplies and scheduling staff. ANs spend about 20% of their time on administrative work, more than half of which is reporting and documentation, while the rest involves updating care plans. AHWs spend only about 10% for their time on administrative tasks, which consist of reports and other documentation.

One essential question is whether any of this work could be transferred to administrative staff. The way in which RNs spend their time today may be seen as a contrast to their educational program, which emphasizes training in patient interaction with the purpose of improving a patient’s physical or mental status. This is supported by other researchers, who have proposed that it is difficult to keep RNs in home services due to the amount of documentation and paperwork that they are required to complete [26,27].

In contrast to RNs, ANs spent a larger proportion of their time on providing practical help and nursing, which together constituted 48% (practical and nursing) of total AN work. Driving accounted for 29% of the total time they dedicated to their work, while documentation and patient administration accounted for 7%. In total, direct care, driving and administration accounts for 85% of the time. Compared to the RNs, ANs used more time on tasks that are directly or indirectly connected to patient work. From this point of view, it seems to be a more rational use of ANs’ skills and training compared to RNs training and skills; however, 15% of the remaining time is spent on other non-patient related tasks.

The AHWs primarily used their time on practical help (50%). The longer median (74 min), as opposed to the corresponding values for ANs (13 min) and RNs (14 min), suggests that the practical help provided by AHWs usually included additional time-consuming tasks. However, they also reported investing 20% of their time in nursing activities which might indicate a shortage of RNs or ANs, thus necessitating the training and use of AHWs in performing simple nursing tasks. In view of the growing demand of home care services and the increasing complexity of home care patient conditions, it seems paradoxical that the members of the AHW category are those who spend the greatest proportion of time with patients.

Today’s home care system in rural municipalities creates working lists by clustering patients in the same geographic area. Such working lists may facilitate efficient systems for municipalities with a widespread geographic area with regard to time spent driving. Based on the current lack of RNs, they could be viewed as a constraint in the future. This means that RNs may be assigned to specific patients based on medical indications. For RNs, such a policy may require more time spent driving, as the patients on such a list are likely to be dispersed over a larger geographic area.

One question that must be addressed is how to utilize and train ANs and AWHs to perform and participate in more advanced medical procedures. In the system today, these two groups are, according to official regulations, not qualified to perform all procedures. However, given the increasing number of patients, new solutions should be considered. It will be important to clarify whether RNs can delegate tasks to ANs and AWHs if they have a certain level of training. This will provide more flexibility in the system. Today, there are concerns about the quality of the care, which may be due to the system and a lack of training or due to workload.

In Norway, home care services have faced significant pressure on technology use. Thus far, Norway has attempted to explore the use of welfare technology in pilot studies. 

Pilot studies in Norway show that the use electronic medication supports a reduced number of home visits (35%) and reduced time per visit (39%). Patients using electronic medication support also report more patient independency [28]. Digital surveillance is also expected to have huge impact on the use of resources. Especially in rural districts in Norway, where the population is widespread and driving distances are significant, digital surveillance will have a tremendous impact in terms of saving resources. One of the municipalities in the pilot reported savings corresponding to one man-year, which is promising [28]. However, this raises ethical questions about how digital surveillance should be used. One may argue that patients will feel safer; others will feel they are being monitored and have no privacy. This becomes especially problematic if the patient has a cognitive impairment. In the future, there must be a discussion of the trade-off between saving resources and intrusions into privacy. Still, many Norwegian municipalities perform manual route planning. Today, there is software available based on optimization algorithm. Optimizing routes may yield up to a 29% decrease in driving time [29]. 

However, more important will be the impact of technology on nursing process performance and the quality of care given, as well as integration with caregivers and patients. These topics should be a priority in future research. More specifically, researchers should prioritize conducting causality analyses and investigating new models for home care services.

Time studies have been used in hospitals to improve quality and increase resource utilization for many decades [30]. However, a time study only provides a partial view of the entire picture. For example, it does not take into consideration patients’ desires, values, family situations and social circumstances, as the person-centered care model does. In a reorganizing processes, these subjects must be taken into consideration. The complexity of unpredicted situations, meaning that tasks may take more time than expected because of patients’ individual needs, must also be considered. This investigation may therefore be used as a starting point for a reorganizing process.

Our research does not explain how to reorganize home care. It only points out how tasks are distributed today. Further investigations may discuss how to reorganize home care in order to use the right resources at the right place on the right patient. We also believe that technology will play a major role in how home care services will organize their work. This will also have an impact on organizing and task distribution in the future.

## 5. Conclusions

The demand for home care services is increasing in terms of both size and complexity. Simultaneously, there is a growing shortage of skilled staff. Our results indicate the existence of major differences in the proportions of the work hours spent on different activities across the three staff categories. 

Regardless of which factor(s) have caused this situation, our research reveals that it is necessary to discuss and modify existing home care service systems in order to use resources appropriately and effectively. It is important to free up health professionals’ time so that patients can receive the services they need. This especially applies to RNs, which was the group with the least face-to-face time with patients. In addition, it will be important to have an increased focus on how to provide more training to ANs and AHW in order for these two groups to perform advanced tasks. To achieve this, the authorities must clarify how responsibilities can be delegated by RNs. This means that RNs may have a larger responsibility in training and delegating to ANs and AHWs.

## Figures and Tables

**Table 1 ijerph-18-10504-t001:** Individual activities and their corresponding clustered activity groups.

Variable	Content
Nursing	Wound management, medical observations, training or rehabilitation, teaching patients and relatives, information, alarm management
Practical help	Hygiene, food facilitating, house cleaning, grocery shopping
Administration organization	Non-specified administration, internal communication, ordering or purchasing, various assignments for public authorities
Administration patient	Staff meetings regarding patients, non-specified patient-related activities, phone calls from patients, relatives and health care staff, pharmacy and laboratory visits
Driving	Traveling to and from patient visits, including food delivery
Reporting	Oral report between work shifts
Teaching	Teaching of employees or students, attending teaching
Documentation (docu)	Writing documentation for each visited patient, adjusting patient care plans
Drug	Multi-dose medication Dispensing

**Table 2 ijerph-18-10504-t002:** Descriptive statistics.

Staff	Proportion	Work	Proportion
	%	Hours	%
AHWs	21	1468.4	22
ANs	40	2783.5	42
RNs	38	2450.3	37
Total	100	6702.2	100

**Table 3 ijerph-18-10504-t003:** The number and proportions of work hours by activity and staff category.

Activity	Work Hours			Proportion of Total (%)		
	RN	AN	AHW	RN	AN	AHW
Practical	28.8	161.7	731.5	1	6	50
Nursing	759.1	1165.1	296.1	31	42	20
Driving	436.6	808.9	251.6	18	29	17
Reporting	254.3	332.6	62.4	10	12	4
Admin org.	331.6	118.8	47.9	14	4	3
Admin patient	191.2	68.8	27.7	8	2	2
Teaching	49.9	21.2	13.9	2	1	1
Documentation	153.7	97.8	22.8	6	4	2
Drug	245.0	8.4	14.4	10	0.3	1
Total	2450.3	2783.5	1468.4	100	100	100

## Data Availability

Data is not available due to confidentiality.

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
