# Peer review of "Who Is Doing What in Home Care Services?"

_ijerph, 2021, doi:10.3390/ijerph181910504_

Round 1

Reviewer 1 Report

Thank you for the opportunity to review this paper, it is an area of interest and ever growing importance and it was interesting to see the distribution and variation in the tasks across the three professions.

Background info

  • It would be helpful to have a description of the patient population - you talk in the discussion about complex patient needs - what proportion typically would this include in the patient population
  • How the home care system is managed - is it national health service provision, are there any costs at point of delivery?
  • How are teams managed, how are workloads distributed across the 3 professions? With reference to the complex case loads - I know you mention this in the discussion, but it would make sense to add to the background
  • What are the differences in pay levels, are there any shortages in staffing?

International context

  • Would be helpful to place Norwegian care system within the international context e.g. there has been a huge debate in the UK in the context of COVID about the undervalue of the social care workforce, with the ANW category of staff one of the lowest paid jobs despite the high value of the work they do. Is this an issue in Norway? Of note, for example, UK ANW workers often do not get paid for travel time between patients.
  • There is also a large shortage in the workforce because of pay and conditions
  • Gendered nature of these job roles 

Methods

It would be helpful to describe how work teams were comprised and how work was allocated and typical workload. 

Discussion

I think there is a fundamental issue about the perceived value of education in performing the home care role. 

"In view of the growing demand of home care services and the increasing complexity of home care patient conditions, it seems paradoxical that the members of the AHW category are those who spend the greatest proportion of time with patients."

  • Are AHWs paid appropriately and valued for the tasks they are undertaking?

Your points about geographical workloads and the use of technology in the discussion are important - does/can your data support this?

Conclusion

Your conclusion suggests that RNs should be spending the most time with patients because they have the highest level of education? I find this quite negatively framed and the issues are far more complex. Just because an RN has more education doesn't mean they should spend more time with patients. I think you need to qualify this statement e.g. spending more 'nursing' time dealing with complex patients. Some of the social interaction-based activities carried out by ANWs and ANs are highly valued.

What are the professional implications for nursing?

What are the implications for the training, pay and conditions for ANs and ANWs?

Author Response

We would like to thank the reviewer for helpful and constructive comments, which have helped us to improve the paper. As requested, we have provided a point-by-point response in the attached document and have also made suggested changes to the paper.

Reviewer 2 Report

There is value in tracking how time is spent by different staff groups within  home care services. The study highlights significant differences between three categories of staff.  It also generates findings which do not correspond with findings of qualitative research on the same topic, suggesting a need for mixed methods. However, there are also risks from tracking tasks and time only within home care services, and the study raises as many questions as it answers. While the authors do already make recommendations for future research, it would improve the paper if the authors were to further highlight the questions that need to be answered in further research. 

It would also improve the paper if the authors were to separate out more clearly 1) amount of time spent driving and the possibility of locality based working to reduce driving time (which might be partly influenced by role) and 2) how time is spent within each home visit (which will also be influenced by role) and 3) amount of time spent on report writing/administration etc (again, role has an influence here). These need to be handled as distinct sub-sections.

It is important not to make assumptions on the basis of counting time spent on tasks. In the UK we have had performance management systems which try to improve efficiency by counting tasks and time.  An excessive focus on task and time in home care is problematic, particularly in undermining person centred care and opportunities to build the skills of service users. When home care is measured and evaluated on the basis of essential task completion, there are missed opportunities for building individual skills and attending to what really matters to the people on the receiving end of home care. Further, home care is a very complex human support system.  There is a great deal of complexity, variety and often unpredictability in the variety of circumstances people are living in and it is important that staff are able to respond to unexpected circumstances in order to improve outcomes for people. The authors already mention this but it is really fundamental to sustainable care which improves outcomes for people - rather than being driven by system priorities focused on task and time only

Author Response

(The authors gave the same response as above.)

Round 2

Reviewer 1 Report

Thank you for taking on board the suggestions, I think you have done a good job improving the paper and happy to recommend for publication. I wish I lived in Norway!

Reviewer 2 Report

This paper is much improved on the previous version.  The authors have responded to my feedback and it seems they have also made changes in response to the other reviewer.  Home care is an increasingly important service and it is useful for countries to be able to learn from each other. While presenting the findings of their own study, this paper also navigates more of the complexity surrounding this important topic and therefore is a useful addition to the evidence base